# Effects of Ketogenic Diet on Muscle Metabolism in Health and Disease

**DOI:** 10.3390/nu14183842

**Published:** 2022-09-16

**Authors:** Elmira I. Yakupova, Alexey D. Bocharnikov, Egor Y. Plotnikov

**Affiliations:** 1A.N. Belozersky Institute of Physico-Chemical Biology, Lomonosov Moscow State University, 119234 Moscow, Russia; 2International School of Medicine of the Future, Sechenov First Moscow State Medical University, 119992 Moscow, Russia; 3V.I. Kulakov National Medical Research Center of Obstetrics, Gynecology, and Perinatology, 117997 Moscow, Russia

**Keywords:** muscle metabolism, ketogenic diet, ketone bodies, fasting, muscle, heart

## Abstract

Dietary intervention is widely used as a therapeutic approach ranging from the treatment of neurological disorders to attempts to extend lifespan. The most important effect of various diets is a change in energy metabolism. Since muscles constitute 40% of total body mass and are one of the major sites of glucose and energy uptake, various diets primarily affect their metabolism, causing both positive and negative changes in physiology and signaling pathways. In this review, we discuss changes in the energy metabolism of muscles under conditions of the low-carbohydrate, high-fat diet/ketogenic diet (KD), fasting, or administration of exogenous ketone bodies, which are all promising approaches to the treatment of various diseases. KD’s main influence on the muscle is expressed through energy metabolism changes, particularly decreased carbohydrate and increased fat oxidation. This affects mitochondrial quantity, oxidative metabolism, antioxidant capacity, and activity of enzymes. The benefits of KD for muscles stay controversial, which could be explained by its different effects on various fiber types, including on muscle fiber-type ratio. The impacts of KD or of its mimetics are largely beneficial but could sometimes induce adverse effects such as cardiac fibrosis.

## 1. Introduction

It is known that muscle is a high energy-demanding tissue of the organism. Sufficient ATP production is essential for muscular contraction and normal muscle functioning. Muscle energetics can affect not only contractile function but also signaling pathways of cell metabolism, growth, survival, and other functions. Changes in energy metabolism can influence muscle tissue not only through ATP supply but also by production of metabolic intermediates.

A feature of muscle energetics is the metabolic flexibility exhibited in response to various stimuli, including changes in consumed food. Muscles can also rearrange their metabolic pathways under pathophysiological conditions, which results in functional shifts. Being one of the main energy consumers in the body and a source of metabolic intermediates, muscles can significantly affect other tissues, playing a role in pathological changes, including metabolic disorders.

In this regard, targeting muscle energy metabolism could be an important therapeutic approach not only for improving muscle functions but also for modulating other tissue functioning in various pathologies. Cardiac pathologies must also be added to the consideration since the heart is also a muscle although somewhat different from skeletal muscles on the cellular and molecular level.

A common way to influence the organism’s energy metabolism is the dietary approach. One of the widespread approaches tested recently as a therapy for diseases associated with metabolic disorders is the low-carbohydrate, high-fat ketogenic diet (KD); another is the dietary restriction or fasting. In this work, we consider the effects of a KD/fasting on muscle physiology in health and disease.

## 2. Muscle Energetics

Skeletal muscles need energy for a large range of activities from maintaining static body position for long periods to performing explosive movements [1], so within a few milliseconds, the energy turnover can change more than 100-fold in response to the full activation from the resting state [2]. Moreover, the mammalian heart must contract incessantly, thus permanently requiring energy [3].

To support muscular contractions, it is necessary to use ATP. The main muscle protein that hydrolyzes ATP is myosin ATPase. ATP is also used for Ca^2+^ uptake into the sarcoplasmic reticulum and maintenance of the sarcolemmal ion gradients. Muscle energy requires a huge accumulation of ATP, while the intracellular ATP content is around 5–6 mM, and this amount depletes within 2 s by fully activated myocytes [2]. There are two main metabolic pathways—anaerobic and aerobic—for ATP recycling. The first is faster and, therefore, dominates during a high-intensity physical activity of short duration. The second is acting during a prolonged submaximal exercise [2]. Anaerobic synthesis of ATP includes degradation of phosphocreatine (PCr): (PCr + ADP ↔ Cr + ATP) and breakdown of muscle glycogen to lactate and hydrogen ions. A minor contribution can also come from myokinase, which is considered to perform a near-equilibrium reaction (2ADP ↔ ATP + AMP) [1].

In adult humans, most glycogen is stored in skeletal muscles (~500 g) and liver (~100 g) [4]. Glycogen is the main energy substrate at the exercise intensity above 70% of the maximal oxygen uptake; fatigue develops when the glycogen stores in the active muscles are depleted [4].

Glycogen is also the major carbohydrate substrate for aerobic metabolism during short-term and prolonged exercise. The contribution of extracellular glucose to oxidative ATP production also increases with the duration of the exercise [5]. During a prolonged submaximal exercise, production of energy is provided by the oxidative metabolism of not only carbohydrates but also lipids [6] as well as amino acids derived from muscle protein degradation [7].

It should be noted that skeletal muscle consists of heterogeneous populations of muscle fibers, which differ in their functions. There are slow-twitch (type I) and fast-twitch (type II) fibers, which contribute to long-term endurance or powerful bursts of movement, respectively. The main differences in these types are their energy metabolisms, especially their reliance on oxidative phosphorylation. Slow-twitch fibers have high mitochondrial content and high oxidative capacity that sustain long-term energy requirements. Fast-twitch fibers have lower mitochondrial content, decreased reliance on oxidative phosphorylation, and include fast-oxidative (type IIa) or fast-glycolytic (type IIb) subtypes [8].

Cardiac muscle, as already mentioned, is somewhat different from skeletal muscle and is well-known as a metabolic omnivore that selects energy substrates suitable for physiological and pathophysiological conditions [3,9,10,11,12]. It is capable of consuming fatty acids, glucose, lactate, ketone bodies (KBs), acetate, and amino acids [3,9,10,11,12]. In the adult heart, the preferred substrate is dynamically changed by substrate availability, hormone effects, oxygen supply, and cardiac workload [13]. Under normal conditions, nearly all ATP is generated by mitochondrial respiration, while 2% or less is derived from anaerobic glycolysis. Fatty acid oxidation (FAO) provides 60–90% of myocardial ATP production in a healthy adult heart, whereas the remaining 10–40% comes from glucose, lactate, amino acids, acetate, and KBs [3,9,10,11,12].

## 3. Metabolism of Ketone Bodies and Its Role in the Muscle

KBs are represented by acetoacetate (AcAc), β-hydroxybutyrate (βHB), and the less abundant breakdown product, acetone. AcAc and βHB are short-chained, four-carbon organic acids produced by the liver, which act as free-fatty acid (FFA)-derived circulating substrates to provide energy to extrahepatic tissues such as the brain, heart, and skeletal muscle, whereas the contribution of acetone, readily generated by spontaneous decarboxylation of AcAc, is negligible for energy provision [14]. There is a balance between hepatic production (“ketogenesis”) of KBs and peripheral breakdown and utilization (“ketolysis”) in extrahepatic tissues in the organism.

KBs are produced in the liver under physiological conditions and after nutritional manipulations that result in reduced carbohydrate (CHO) availability, including prolonged fasting, starvation, and ketogenic diet [15,16]. Thus, under normal conditions, the metabolism of ketone bodies in muscle (ketolysis) cannot start without a ketogenesis in the liver (Figure 1).

The increase in KB levels leads to attenuating glucose utilization in peripheral tissues, anti-lipolytic effects in adipose tissue, and potential attenuation of proteolysis in skeletal muscles [14,15]. KBs are utilized by working muscle during exercise [17,18], and the capacity to take up and oxidize them in exercise is higher in exercise-trained skeletal muscle [19].

The activity of the enzymes involved in ketone utilization is different depending on muscle fiber types. BDH, OXCT, and ACAT had the highest activity in type I fibers; moderate in type IIA; and lowest in type IIB fibers of rat muscle [20]. In all fiber types, the BDH activity is much lower than the activities of OXCT and ACAT. Moreover, the activity of the enzymes is dependent on the muscle training status: in adaptation to exercise the BDH activity increases 3-fold in type I and 6-fold in type IIA fibers [20]. After a 14-week-long program of treadmill running, OXCT and ACAT increased their activity as well. It was 26% higher in type I and approximately two times higher in type IIA and IIB fibers for OXCT. The ACAT activity was 40–45% higher in all three fiber types [20]. There are no changes in these enzymes in the heart after training [20].

Expression of ketolytic enzyme genes in skeletal muscle was also altered after training. BDH mRNA expression was 2-fold higher, while OXCT and ACAT mRNA expression was 30–50% higher [21]. Moreover, PGC-1α was identified as a transcriptional regulator of ketolytic enzymes and KB transporters in skeletal muscle [21]. It was shown that muscle-specific overexpression of PGC-1α can reduce hyperketonemia in both healthy and diabetic mice [21]. Svensson et al. also showed that modulation of ketolytic gene transcription in skeletal muscle by PGC-1α affects systemic ketosis in response to various stimuli such as not only exercise but also fasting and low-carbohydrate diet/ketogenic diet.

## 4. Ketogenic Diet

The history of ketogenic diet started in 1921 when Woodyatt discovered that acetone and beta-hydroxybutyric acid were produced in healthy humans during starvation or a diet containing a low proportion of carbohydrates and high proportion of fat [22]. The term “ketogenic diet” was coined by Wilder, who investigated how the diet influences epilepsy patients. Then, Peterman (1925) described KD similar to that used today: 1 g of protein per kilogram of body weight, 10–15 g of carbohydrates per day, and the remainder of the calories in fat. Thus, the standard KD consists of four parts fat (long-chain triglycerides) to one part combined protein and carbohydrate ratio, supplemented with vitamins and minerals [23]. It is known that KD is a highly effective treatment for medically intractable epilepsy and has been applied in clinics for over 70 years now [24,25]. To date, the beneficial effects of KD for treating Alzheimer’s disease [26], Parkinson’s disease [27], brain injury [27,28,29], amyotrophic lateral sclerosis [30,31,32], brain trauma (concussion) [33], migraine [34], polycystic ovary syndrome [35], and cancer [36] have been shown.

Ketone body synthesis, or ketogenesis, takes place mostly in hepatocytes and to a lesser extent in astrocytes or kidney cells [37]. Main KB utilization occurs in the heart, skeletal muscles, and brain [37,38,39]. Most of the effects of KD are associated with changes in organism metabolism. Due to the lack of or reduction in dietary carbohydrates, a decrease of insulin and increase of glucagon in plasma are observed, which promotes hepatic glycogenolysis and gluconeogenesis as well as lipolysis of adipose tissue through an increase of hormone-sensitive lipase. Restriction of carbohydrates over 4–7 days leads to an exhaustion of glycogenolysis and an increase of ketogenesis accompanied by a rise of free fatty acids (FFA), acetyl-CoA (through mitochondrial beta-oxidation), and KB levels [40,41] (Figure 2). Under conditions of the low levels of glucose or the lack of carbohydrate consumption, KBs are the predominant energy source when fat is metabolized while mediating physiological processes such as cell signaling, post-translational modifications, inflammation, oxidative stress, and synthesis of lipids such as myelin and cholesterol [42].

KD may result in ketosis characterized by ketonemia with maximum KB levels of 7–8 mmol/L (and not going higher because the central nervous system efficiently uses these molecules for energy instead of glucose) and without any change in pH [44].

There is a pathogenic state called ketoacidosis that develops when ketonemia exceeds 20 mmol/L with a concomitant lowering of blood pH [44]. This state can be lethal. Thus, there are two states: “physiological ketosis” and lethal ketoacidosis [24].

Since KD is used conventionally for the treatment of refractory pediatric epilepsies, discussing the mechanism of KD neuroprotective effects is more common in the literature [45]. However, recently interactions of KBs and ketosis with muscle, especially cardiac muscle, started to draw more and more attention [46,47,48].

## 5. Ketone Bodies Production during Fasting

The KB content could be increased not only by KD but also by fasting.

Normally, KBs are produced by the human liver in the amount of about 300 g per day, and ketones could provide approximately 5% of the energy requirements in the fed state, rising to 20% in long-term fasting [38,49]. Thus, prolonged fasting or vigorous exercise may lead to an excess of ketones and cause ketosis [40]. During fasting, hepatic reserves of glycogen are first broken down by glycogenolysis to deliver glucose to the circulation. After 1–2 days of fasting, glucose is largely available through gluconeogenesis, while afterwards, the glycogen stores are depleted, and gluconeogenesis works together with glycogenolysis to meet the energy requirements of the organism. In this case, the main substrate for gluconeogenesis is 3-monoacylglycerol [40,50], which is produced by hydrolysis of triglycerides in adipose tissue. During fasting, the liver preferentially converts fatty acids to ketone bodies. This metabolic switch occurs at the level of hepatic mitochondria [51] since it is the major site of ketone body synthesis and β-oxidation.

Under physiological conditions, the blood KB concentration ranges in humans from 0.05 to 0.1 mM and could rise beyond 0.5 mM to reach even 5–7 mM during prolonged fasting [52], starvation, caloric restriction, ketogenic diet [15,16], and exercise [53].

## 6. Ketogenic Diet and Skeletal Muscle

To date, a large body of data has been accumulated about the impact of KD on various muscles (Table 1). It has been shown to reduce muscle weight, fiber area, and grip strength [54] by upregulation of muscle atrophy-related genes Mafbx, Murf1, Foxo3, Lc3b, and Klf15 [54]. Thus, KD can lead to muscle atrophy in which hypercorticosteronemia, hypoinsulinemia, reduced insulin-like growth factor 1 (IGF-1), and oxidative stress are involved [54]. However, there are studies where KD increases body weight and fat mass [55,56,57] or does not change body mass and muscle mass [58].

Certainly, the main KD influence on the muscle is reduced to changes in energy metabolism. The circulating levels of FFA increased up to 700% in mice fed with KD compared to control. The activity of pyruvate dehydrogenase (PDH), which catalyzes the oxidative decarboxylation of pyruvate into acetyl-CoA and links glycolysis to the citric acid cycle, was significantly decreased in skeletal muscles of KD-fed mice [54]. Along with this, the expression of pyruvate dehydrogenase kinase 4 (PDK4), which contributes to PDH phosphorylation and impaired glucose utilization, also increased 2.2-, 2.8-, and 3.8-fold in skeletal muscles (gastrocnemius, tibialis anterior, and soleus, respectively) after KD [54]. An increase in PDK4 content in skeletal muscle was repeatedly confirmed after KD [59,60] as well as KD-induced decline in carbohydrate utilization capacity during exercise [61,62,63], indicating that muscle tissue after KD shifted the preferred energy substrate from glucose to fat. It was shown that KD with a medium-chain triglyceride (MCT) content increased the ketolytic capacity in skeletal muscle without exerting inhibitory effects on carbohydrate metabolism [64] in contrast to long-chain triglyceride (LCT) content.

Wallace et al. (2021) showed the effectiveness of a long-term KD in mitigating sarcopenia, the muscle disease associated with reduced muscle strength and metabolic abnormalities in aging [65]. The influence on muscle weight differed between different muscles (see Table 1). Beneficial effects of KD were associated with a shift in fiber type from type IIb to IIa fibers, increased markers of neuromuscular junction remodeling, mitochondrial biogenesis, oxidative metabolism, and antioxidant capacity while decreasing endoplasmic reticulum stress, protein synthesis, and proteasome activity [65].

KD could also affect the mitochondrial structure and function in skeletal muscle (Table 2). Oxidative stress defense and improved mitochondrial quality are potential mechanisms through which KD may provide physiological benefits [66,67]. Some studies showed an increase in markers of mitochondrial biogenesis after KD [65], while others demonstrated it not to change mitochondrial content [59]; but KD with exercise training showed increases in markers of mitochondrial fission/fusion [55]. Furthermore, KD was found to enhance mitochondrial respiration without increased mitochondrial content in skeletal muscle [56,68]. Transmission electron microscopy analysis, however, showed that both subsarcolemmal and intermyofibrillar fractions were significantly higher in KD mice accompanied by an increase in mitochondrial content in both regions [69].

One can see contradictory data about mitochondrial quantity after KD impact (Table 2). Indirect estimation of the mitochondrial content by measuring the maximal citrate synthase activity [68] found a lowering of mitochondrial quantity in KD-fed mice. Other studies demonstrated no changes in mitochondrial quantity [55,56,59] nor its increase [69] after KD. This latter effect confirms the previous data about an increase of mitochondrial biogenesis by KD [65]. Nevertheless, it was also shown that mitochondrial ROS production was increased, mitochondrial glutathione decreased and gastrocnemius pyruvate-malate mitochondrial respiratory control ratio was impaired after KD [68].

There is another piece of contradictory data: (1) a ketogenic diet increases citrate synthase (CS) activity and improves oxidative capacity in extensor digitorum longus skeletal muscle [70]; (2) it lowers mitochondrial function [68,71] and decreases CS activity in gastrocnemius [68] after KD. An explanation could be the differences in slow-twitch fiber content in the muscles. Thus, Parker et al. reports that a four-week KD decreases CS activity in the red portion but not in the white portion of gastrocnemius muscles in rats [56]. In turn, Ogura et al. notes that muscle fiber types need first to be studied to ascertain KD effects on the enzymes in skeletal muscle [70]. This assumption can be applied to all research of KD–muscle crosstalk.

One can see different effects of KD on skeletal muscle that can be partially explained by fiber type, while KD itself can lead to a change in the fiber type ratio in the muscle. However, in all cases, we can assume changes in energy metabolism, in particular decreased carbohydrate oxidation and increased fat oxidation. The problem lies in contradictory data from studies of KD effects, starting from its influence on muscle weight to mitochondrial quantity, oxidative metabolism, antioxidant capacity, and activity of enzymes.

**Table 1 nutrients-14-03842-t001:** Effects of KD on skeletal muscles.

Animal	KD: Protein:Fat:Carbohydrate Ratio (% Total Energy)	Muscle	Exposure/Pathology	Outcome	Ref.
Male C57BL/6 J mice	KD: 19:61:20	*Extensor digitorum longus*, *soleus*, *gastrocnemius*, and *quadriceps femoris*	Physically active mice had access to a running wheel	KD-fed mice were more insulin-resistant. KD led to upregulation of PDK4 and ERRα mRNA and protein levels. Effects of KD with running were even greater. PGC-1α mRNA or protein levels did not significantly depend on KD. The amount or function of mitochondria did not depend on KD either.	[59]
Seven-week- old male Sprague–Dawley rats	LKD: (12:87:1); MKD: (16:66:18)	*Epitrochlearis* and *triceps*	Swimming exercise (8-week intervention)	Endurance training significantly increased OXCT content in *epitrochlearis* muscle tissue and additionally increased OXCT protein content. LCT but not MCT diet substantially increased muscle PDK4 protein level. Thus, MCT diet may additively enhance endurance training-induced increases in ketolytic capacity in skeletal muscle without exerting inhibitory effects on carbohydrate metabolism.	[64]
Six-week-old female Jcl:ICR mice	KD: 4.8:94.8:0.1	*Gastrocnemius*, *tibialis anterior*, and *soleus*	-	Circulating levels of FFA increased up to 700% in KD-fed mice. KD increased plasma corticosterone levels 2.9-fold and decreased plasma IGF-1 levels by 60% in comparison. Feeding mice with KD led to upregulation of muscle atrophy-related genes *Mafbx*, *Murf1*, *Foxo3*, *Lc3b*, and *Klf15* in skeletal muscles. Expression of anabolic genes such as Igf1 and Col1a2 is reduced in KD group. mRNA expression of oxidative stress-responsive genes such as Sod1 was significantly increased after KD. Thus, KD can lead to muscle atrophy in which hypercorticosteronemia, hypoinsulinemia, reduced insulin-like growth factor 1 (IGF-1), and oxidative stress are involved.	[54]
C57BL/6 mice	KD: 10:89:1	*Quadriceps* (QUAD), *gastrocnemius* (GTN), *plantaris* (PLN), *soleus* (SOL), *tibialis anterior* (TA), and *extensor digitorum longus* (EDL)	Age-related sarcopenia	No differences in body weight were observed with age or diet. The influence on muscle weight was different between different muscles: GTN significantly decreased in weight from 16 to 26 months after KD and in control. After 26 months, GTN was significantly bigger in KD mice; PLN and SOL had a significantly lower muscle mass after 26 months compared to 16, and there was a trend for higher PLN and SOL muscle weights in 26-month-old mice on KD compared with 26-month-old control. This result in KD mice was associated with a shift in fiber type from IIb to IIa fibers and alterations in a range of molecular parameters, including increased markers of neuromuscular junction remodeling, mitochondrial biogenesis, oxidative metabolism, antioxidant capacity, as well as decreased endoplasmic reticulum stress, protein synthesis, and proteasome activity.	[65]
C57BL/6J male mice	KD: 16.1:83.9:0	*Quadriceps*, *gastrocnemius*	Exercise training (ExTr): Mice ran on a treadmill for 3 weeks, 5 days/week, 1 h/day	ExTr increased intramuscular glycogen, whereas KD increased intramuscular triglycerides. Neither KD nor ExTr alone altered mitochondrial content; however, in combination, the KD–ExTr group showed an increase in PGC-1α and markers of mitochondrial fission/fusion. There are no changes in pyruvate oxidative capacity by both exposure. KD and ExTr interventions both enhanced mitochondrial and peroxisomal lipid oxidation and adaptations were additive or synergistic.	[55]
C57BL/6 mice	KD: 9.9:74.4:3	*Quadriceps*	PGC-1α mKO	KD increased oxygen consumption by muscle in PGC-1α-dependent manner, concomitant with a blunted transcriptional induction of genes involved in fatty acid oxidation and impairment in exercise performance.	[71]
Sedentary Wistar rats	KD: 10:90:0	*Extensor digitorum longus*	-	Physiological parameters such as twitch or tetanic forces or muscle fatigue did not differ between KD and the control group. Citrate synthase activity and protein levels of Sema3A, citrate synthase, succinate dehydrogenase, cytochrome c oxidase subunit 4, and 3-hydroxyacyl-CoA dehydrogenase were significantly higher in KD group. The 4-week ketogenic diet improves skeletal muscle aerobic capacity.	[70]
Male Fisher 344 rats	KD: 22.4: 77.1:0.5	White and red *quadriceps*		KD induces weight loss and enhances mitochondrial respiration without increased mitochondrial content in skeletal muscle. Muscle after KD similarly produced less H_2_O_2_ despite an increase in mitochondrial respiration and no apparent change in mitochondrial quantity.	[56]
Male Fisher 344 rats	KD: 23:67:10	*Gastrocnemius*		KD-fed rats have significantly greater mitochondrial ROS production in the gastrocnemius, while mitochondrial glutathione levels were lower. The GTN pyruvate-malate mitochondrial respiratory control was significantly impaired in long-term KD-fed rats, and GTN mitochondrial quantity was lowest in these animals. Rotarod performance was greatest in KD-fed rats.	[68]
C57BL6 mice	KD: 8.6:75.1:3.2	*Tibialis anterior*	BaCl2 injury	Muscle stem cells (MuSCs) isolated from 3-week KD-fed mice demonstrate a deep quiescent (DQ) state. This state is characterized functionally and transcriptionally as being less committed to a myogenic program but also enhanced resistance to nutrient, cytotoxic, and proliferative stress. Impairment of regeneration of muscle tibialis anterior was shown, when mice had been fasted for 2.5 days and subsequently refed for 1, 2, 3, or 7 days prior to BaCl2 injury.	[72]
C57Bl/6J mice	KD: diets containing 5% sodium butyrate (butyrate) for 8–10 months	*Gastrocnemius–plantaris* with *soleus*	Age-related muscle atrophy	KD abolished age-dependent muscle mass loss, and the mice treated with butyrate from 16 months old were protected against muscle atrophy in hindlimb muscles. Diet containing butyrate increased muscle fiber cross-sectional area, prevented intramuscular fat accumulation in the old mice, and improved glucose metabolism in 26-month-old mice as determined by a glucose-tolerance test. The effect of butyrate was not associated with reduced ubiquitin-mediated proteasomal degradation. Butyrate reduced markers of oxidative stress and apoptosis and improved antioxidant enzyme activity. It is supposed that these effects were associated with HDACs inhibition by butyrate.	[73]
C57BL/6J mice	KD: 20:69:10; 20:78:1	*Gastrocnemius*	Small animal treadmill set to 15 m/min, 0% slope, for 60 min for 5 days	No change of body and muscle mass was observed in two groups of animals after KD with 1% and 10% of protein content compared to control. Only in the KD group with 1% of protein content did the plasma ketone concentration significantly increase with gene expression related to glucose utilization significantly declining in the muscle. In both groups, an increased gene expression related to lipid utilization was observed. Thus, KD with 10% of proteins leads to an increase of lipid oxidation without ketosis and suppression of muscle glucose utilization. However, in both groups, KD treatment did not affect endurance capacity.	[58]
Male Sprague–Dawley rats	KD: 20.2:69.5:10.3	*Gastrocnemius*	Cage with a resistance-loaded voluntary running wheel	In the presence of a complex II (succinate) substrate, the respiratory control ratio of isolated gastrocnemius mitochondria was higher in animals fed the KD. Complex I (pyruvate + malate) and IV enzyme activity was higher in EXE (exercised using resistance-loaded running wheels) animals regardless of diet. SOD2 protein levels and GLUT4 and PGC1α mRNA expression were higher in EXE animals regardless of diet.	[67]
Male C57BL/6JN mice	KD: 10:89:1	*Gastrocnemius*		Both subsarcolemmal and intermyofibrillar fractional area was significantly higher in KD mice, consistent with an increase in mitochondrial content in both regions. There was no difference in the average size of mitochondria between diet groups.	[69]

A meta-analysis of KD effects on the body composition and muscle performance with exercise endurance in humans with a particular focus on randomized controlled trials did not clearly support a major beneficial effect of KD on the muscle/fat ratio or performance in strength-trained individuals. Thus, it is too early today to make any conclusions about the efficiency of KD in sports [74].

**Table 2 nutrients-14-03842-t002:** Influence of KD on mitochondrial content and function in skeletal muscle.

Animal	KD	Muscle	Method of Analysis	Mitochondrial Function		Ref.
Male C57BL/6 J mice	KD: 19:61:20	*Extensor digitorum longus*, *soleus*, *gastrocnemius*, and *quadriceps femoris*	Electron microscopy analysis of mitochondrial content; respiration rates procedure	No change		[59]
Male Fisher 344 rats	KD: 23:67:10	*Gastrocnemius*	Maximal citrate synthase activity; respiration assays; mitochondrial ROS determination; tissue mitochondrial glutathione assays	↓	Mitochondrial ROS production ↑ Mitochondrial glutathione ↓ Gastrocnemius pyruvate-malate mitochondrial respiratory control ratio (impaired)	[68]
Male Fisher 344 rats	KD: 22.4: 77.1:0.5	White and red *quadriceps*	Respiration rates procedure; Western blotting of cytochrome c	No change	Mitochondrial respiration ↑	[56]
C57BL/6 mice	KD: 10:89:1	*Gastrocnemius*	Expression and levels of several transcriptional regulators of mitochondrial biogenesis	-	Mitochondrial biogenesis ↑	[65]
C57BL/6J male mice	KD: 16.1:83.9:0	*Quadriceps*, *gastrocnemius*	Estimation of substrate oxidation rates	no change	-	[55]
Male C57BL/6JN mice	KD: 10:89:1	*Gastrocnemius*	Transmission electron microscopy	↑	-	[69]

↑ Increase; ↓ Decrease.

## 7. Fasting and Skeletal Muscle

During starvation, KBs become the major source of energy for several tissues, including skeletal muscles. They are found to provide about 51% of the O2 consumption of resting forearm muscle after 84 h of [75]. However, in forearm catheterization studies with starvation on lean individuals, ketone bodies were metabolized in the following proportions: 5% after an overnight fast; 10% after 30–36 h of starvation; and 20% after 60–66 h of starvation [76]. The differences can be associated with the choice of patients: in [75], data were obtained from obese subjects, while in [76], data were from lean individuals. The study on lean subjects also suggests that fat rather than ketone bodies is the dominant fuel for muscle during short-term starvation [76]. The effect of fasting directly depends on the effect of KBs on skeletal muscles.

## 8. Effect of KBs on Skeletal Muscles

Effect of KD/fasting is tightly associated with KBs. Apart from acting as an energy substrate, they may work as signaling molecules to regulate gene expression and adaptive responses [77,78] (Figure 3).

Shimazu et al. showed that βHB is an endogenous and specific inhibitor of class I histone deacetylases (HDACs). Using a cell model, researchers demonstrated increasing histone acetylation at the Foxo3a and Mt2 promoters after treatment with βHB. Both genes were activated by selective depletion of HDAC1 and HDAC2 [77]. Increased FOXO3A and MT2 activity after treatment of mice with βHB provided substantial protection against oxidative stress, reduced muscle atrophy, prevented intramuscular fat accumulation, and elevated oxidative metabolism during aging [77].

It is known that short-term fasting is beneficial for the regeneration of multiple tissue types [79,80]. While ketosis provoked by fasting or a ketogenic diet led to a deep quiescent state of the MuSCs, slowing muscle regeneration [72], these MuSCs demonstrated increased survival and enhanced resilience to nutrient, cytotoxic, and proliferative stress. The mechanism of this effect can be due to β-hydroxybutyrate functions as an HDAC inhibitor in MuSCs, leading to acetylation and activation of an HDAC1 target protein p53. This MuSCs state, as it has been shown, is dependent on p53 activation [72]. A study on mice showed an impairment of regeneration after BaCl2 injury persisted up to 3 days following a refeeding after 2.5 days of starvation [72].

Interestingly, KD leads to upregulation of muscle atrophy-related genes Mafbx, Murf1, Foxo3, Lc3b, and Klf15 [54] that can cause muscle atrophy [54]. Thus, today, the question about the adverse or protective effect of βHB in muscle dystrophy stays open. 

Moreover, βHB precursor, AcAc, showed a functional role in muscle regeneration and ameliorates muscular dystrophy in muscular dystrophin-deficient mice model. That is possible due to AcAc stimulated myoblast proliferation by up-regulating cyclin D1 expression through activation of the MEK1-ERK1/2 signaling cascade [78].

## 9. Ketogenic Diet/Fasting/KBs and Cardiac Muscle

The heart is an omnivore with respect to energy substrates, but the utilization of βHB by it is low. However, KB metabolism is important for the heart adaptation to pathological conditions, which was shown on various mouse models with specific OXCT knock-out or a cardiac-specific BDH1 overexpression. The OXCT knock-out resulted in significantly increased left ventricular volume and decreased left ventricular ejection fraction during pressure overload by transverse aortic constriction (TAC). This was accompanied by an increase in the myocardial reactive oxygen species, mitochondrial damage, and disruption of myofilament ultra-structure [81,82]. Vice versa mice with BDH1 overexpression were resistant to pathological cardiac remodeling under the stress of TAC, with limited impairment of cardiac function and attenuated cardiac fibrosis and hypertrophy. Thus, protective effects of KB utilization by the heart were demonstrated due to a decrease of ROS-induced DNA damage as well as improved mitochondrial ROS generation and apoptosis in failing hearts [82]. In addition, after TAC and myocardial infarction surgery the expression of BDH1 in failing hearts was significantly upregulated, which can indirectly indicate an importance of KB utilization in the heart adaptation [83]. Beneficial effects of therapy with ketone body, 3-hydroxybutyrate, was shown in clinical study for treatment patients with heart failure [84]. Infusion of 3-hydroxybutyrate in patients with heart failure led to increased cardiac output by 2 L/min (40%) with an improvement in left ventricular ejection fraction (8%) [84].

We can also note that therapeutic efficiency of some cardioprotective agents, for instance sodium glucose transporter 2 inhibitors (SGLT2i), can be associated with ketogenesis stimulation [85,86,87]. In a porcine model of heart failure one of SGLT2i, empagliflozin, led to switch of metabolism from glucose toward ketone bodies. Empagliflozin-induced ketogenesis was associated with amelioration of cardiac remodeling and heart failure in a nondiabetic model cardiac disfunction [87].

Using the model of heart failure associated with the mitochondrial pyruvate carrier (MPC) deletion, McCommis et al. showed protective effects of KD. Pyruvate, in order to enter the mitochondrial matrix and to be oxidized, must be transported across the inner mitochondrial membrane by the MPC [88,89], and genetic deletion of the MPC in mice led to cardiac remodeling and dysfunction, while KD could prevent or even reverse this heart failure [90]. Fasting during 24 h also provided a significant improvement in heart remodeling [90]. The mechanism of these improvements could be based on a KD increase of cardiac fat oxidation and limitation of carbohydrate provision but without enhancing the ketone metabolism [90].

The cardio-protective effects of KD were also shown in a global ischemic injury model in rats [47], where 19-week KD improved the recovery of coronary blood flow and increased the number of mitochondria in the heart [47].

Administration of high concentrations of dl-3-hydroxybutyrate (DL-3-HB) and D-β-hydroxybutyrate also protected mouse and rat hearts against the coronary artery occlusion injury model [91,92]. The DL-3-HB treatment reduced myocardial infarction size and apoptosis, possibly by providing increased energy substrate to the fasted rat myocardium [92]. βHB promoted autophagic flux [92], as shown by the reduced ratio of LC3-II/LC3-I, decreased levels of p62 and increased lysosome associated membrane protein-2 (Lamp2) in myocardium [92]. Furthermore, the treatment of mice with βHB decreased oxidative stress and attenuated endoplasmic reticulum stress in the ischemic heart [92]. Increasing cardiac ketone delivery by chronic oral ketone ester (KE) supplementation ameliorates cardiac dysfunction in transverse aortic myocardial infarction (MI) model in mice and post-MI remodeling model in rats [93]. The cardioprotective role of βHB was also shown in tachycardia-induced myopathy [94]. β-Hydroxybutyrate could directly ameliorate inflammation via reduced NLRP3 (nucleotide-binding domain-like receptor protein 3)-inflammasome activation [95], thus providing cardioprotective effects during the heart failure [96].

Surprisingly, in another study, the treatment with βHB exacerbated cardiomyocyte death decreased glucose absorption and glycolysis under hypoxic conditions [97]. Moreover, KD was followed by aggravated cardiac dysfunction in MI mouse models [97,98], and this effect was ameliorated by inhibiting hypoxia-inducible factor 1α (HIF-1α) degradation [97]. Similarly, the study of isolated hearts showed that KD increased myocardial injury following ischemia/reperfusion (I/R) [99], resulting in impaired left ventricular performance, reduced recovery, and 10- to 20-fold increased injury as measured by lactate dehydrogenase release and histologic infarct area [99]. Indeed, 3-day fasting increased the concentration of βHB and βHB/acetoacetate ratio that could induce protection of rat hearts against acute I/R injury [100]. The fasting limited myocardial infarct size and reduced the occurrence of premature ventricular complexes as well as reperfusion-induced ventricular arrhythmias after coronary artery occlusion [100].

We can conclude that KD or increased βHB bring beneficial effects in heart failure and I/R injury (Figure 4).

In addition to the described negative phenomena, a 4-week KD could lead to aggravating interstitial fibrosis and cardiac remodeling in spontaneously hypertensive rats (SHR) [101] and stimulated the development of hypertension in SHR [102]. Expectedly, βHB strengthened the progression of TGF-β-induced fibrosis in isolated cardiac fibroblasts. These effects could be mediated by the activation of the mTOR pathway, and suppressing mTOR is a target for preventing hypertension and its related fibrosis [101]. 

Xu et al. also reported that prolonged KD exposure induced cardiac fibrosis. This effect was promoted by the HDAC2 inhibitor effect of βHB that in turn caused histone acetylation of the SIRT7 promoter and activated SIRT7 transcription [103]. SIRT7 inhibited the transcription of mitochondrial ribosome-encoding genes and mitochondrial biogenesis, leading to cardiomyocyte apoptosis and cardiac fibrosis [103]. The same effects of KD were shown in obese rats [104].

Thus, despite of beneficial effect on cardiac muscle, there are few studies that demonstrate detrimental effects on the heart, especially in a prolonged exposure; these adverse events include hypertension and cardiac fibrosis (Figure 5).

## 10. Conclusions

KD effects on the skeletal and cardiac muscle are contradictory with respect to muscle weight, mitochondrial content, oxidative metabolism, antioxidant capacity, activity of various enzymes, as well as an impact on some pathological states.

Although many studies demonstrate beneficial effects of KD and fasting on skeletal or cardiac muscle, these diets should be used carefully because there are data suggesting that such nutritional manipulations can have negative, including long-term, consequences.

## Figures and Tables

**Figure 1 nutrients-14-03842-f001:**
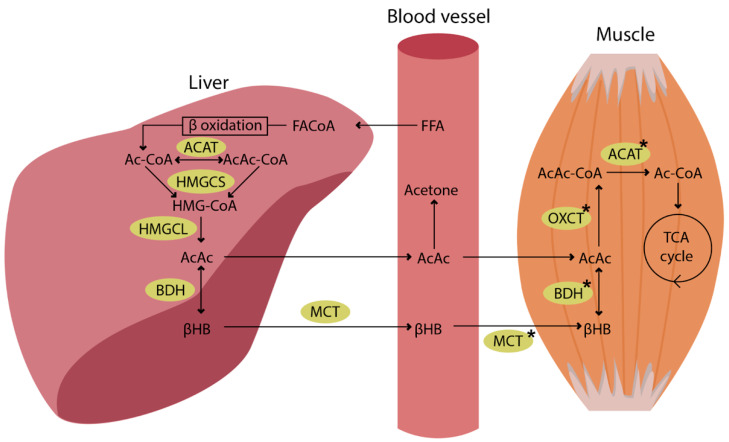
Ketogenesis in the liver and ketolysis in the muscle (for details see [14]). FFAs are transported to the liver and converted to fatty acyl CoA (FA-CoA). The next step after β-oxidation is condensation of Ac-CoA molecules to acetoacetyl CoA (AcAc-CoA) by mitochondrial thiolase activity of Ac-CoA acetyltransferase (ACAT). Then, in a reaction with hydroxymethylglutaryl CoA synthase (HMGCS) generates hydroxymethylglutaryl-CoA (HMG-CoA). HMG-CoA decomposes to AcAc by HMG-CoA lyase (HMGCL). AcAc can be released to the circulation, but most of it is reduced to βHB by 3-hydroxybutyrate dehydrogenase (BDH). βHB can undergo interconversion with AcAc in the liver and in other tissues after its uptake from the blood. Succinyl-CoA:3-oxoacid CoA transferase (OXCT) catalyses the generation of AcAc-CoA from AcAc and succinyl-CoA. Finally, AcAc-CoA is cleaved to Ac-CoA by ACAT, and Ac-CoA is incorporated into the TCA cycle. The asterisk (*) indicates protein content and enzyme activity that are higher in the exercise-trained skeletal muscle.

**Figure 2 nutrients-14-03842-f002:**
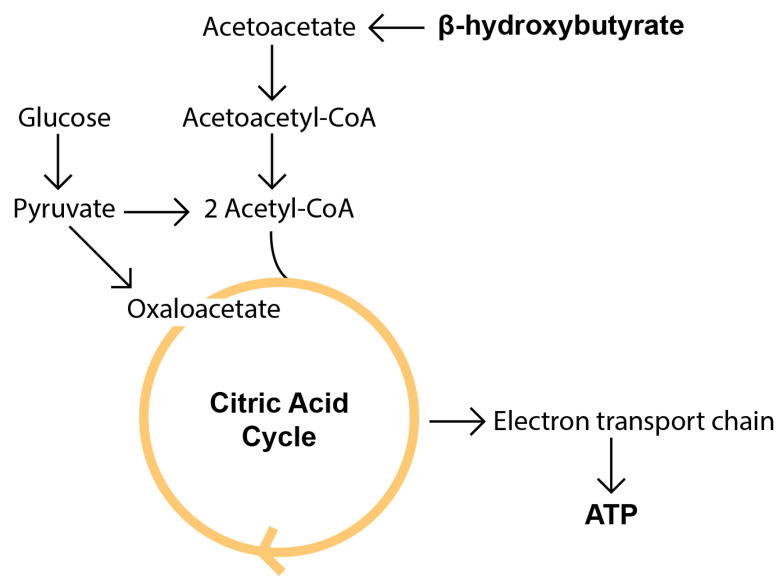
The basic metabolism of beta hydroxybutyrate compared to glucose. KBs arise from liver fatty acid metabolism occurring primarily in the hepatic mitochondrial matrix. Then, they are captured from circulation by extrahepatic tissues and used as an energy source. βHB is oxidized by βHB dehydrogenase to produce AcAc, which takes up CoA from succinyl-CoA to produce acetoacetyl-CoA. Acetoacetyl-CoA then reacts with CoA to produce two molecules of acetyl-CoA, a key intermediate for the citric acid cycle (reviewed in [16,37,43]).

**Figure 3 nutrients-14-03842-f003:**
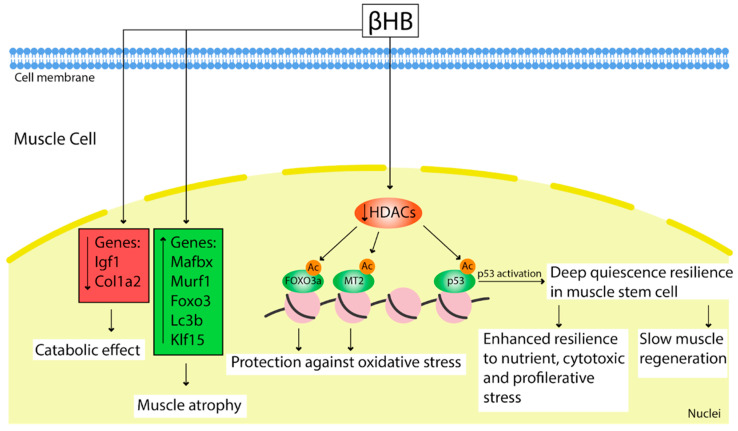
Effect of elevated βHB in muscle cells after KD/starvation or exogenous administration. βHB caused upregulation of muscle atrophy-related genes Mafbx, Murf1, Foxo3, Lc3b, and Klf15 and reduction in expression of anabolic genes such as Igf1 and Col1a2. As an HDACs inhibitor, βHB promotes acetylation of FOXO3a and MT2 promoters, providing substantial protection against oxidative stress. Inhibition of HDACs also activates the p53 protein that leads to a deep quiescence resilience state in muscle stem cells. This state is associated with slow muscle regeneration and enhanced resistance to various cell stresses.

**Figure 4 nutrients-14-03842-f004:**
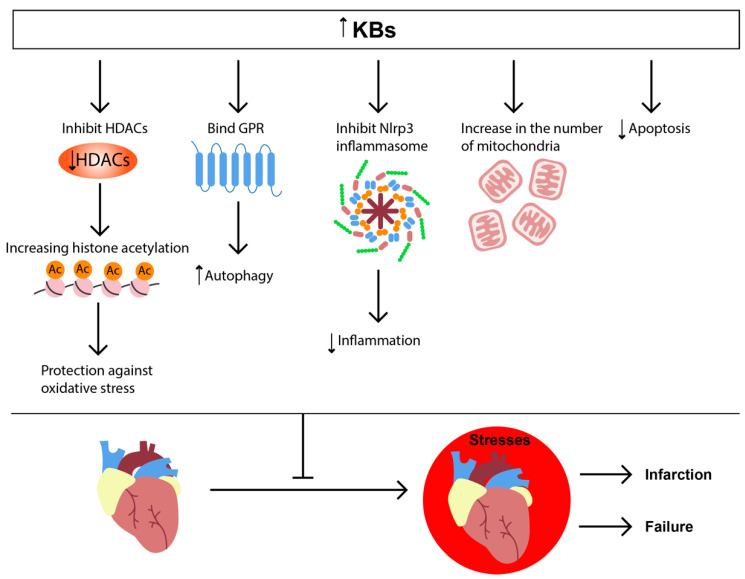
KB signaling in heart pathologies. The increased concentration of KBs leads to binding with GPR, inhibition of HDAC and NLRP3-inflammasome, increase in the number of mitochondria, and decrease of apoptosis. These effects protect the heart by enhancing autophagy and reducing inflammation and ROS production.

**Figure 5 nutrients-14-03842-f005:**
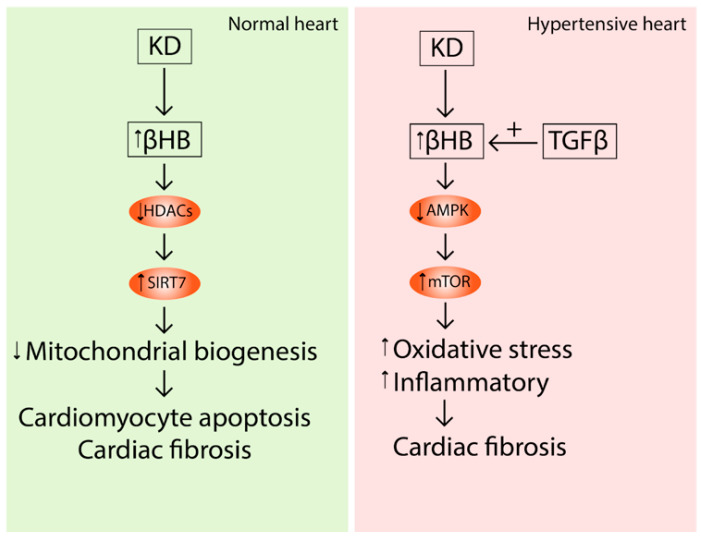
Long-term effects of KD and increased levels of βHB in normal and hypertensive hearts. βHB inhibits the HDAC2 that activates Sirt7 transcription leading to cardiomyocyte apoptosis and cardiac fibrosis in wild-type rats. In hypertensive hearts, KBs inhibit AMPK signaling, activate mTOR and result in cardiac fibrosis. This effect can be stimulated by TGF-β. (↑) Activation; (↓) Inhibition.

## Data Availability

Not applicable.

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
