# Peer review of "Effects of Ketogenic Diet on Muscle Metabolism in Health and Disease"

_nutrients, 2022, doi:10.3390/nu14183842_

Round 1

Reviewer 1 Report

I read with great interest the paper “Effects of ketogenic diet on muscle metabolism in health and 2 disease" by Yakupova et al.

The review is well written. Paper design is fine. The article is logically divided into sections and subsections.

Comment:

1.      I would give more prominence to the role of ketone bodies in the heart in section 2 as they may represent a benefit in some structural disease such as heart failure. In fact, ketone bodies represent a good alternative substrate, able to improve the cardiac metabolic efficiency. Some studies on humans and animal models have shown an improvement of the cardiac function and metabolism by beta-hydroxybutyrate (β-OHB), thus inducing reverse ventricular remodelling, with a consequent improvement in the cardiac output and diastolic function. (doi: 10.3390/ijms22115863).

Author Response

Dear Editors, 

Thank you for giving us a chance to revise our manuscript. We appreciate the work of the editor and reviewers and express great gratitude for the positive assessment of our article. We have modified the text to answer all recommendations expressed in the reviewer’s comments. We have made the appropriate changes in the text.

Thank you for your work.

Below, we present the reviewer’s specific comments (in black) with our replies (in blue).

Sincerely,

Elmira I. Yakupova,
Egor Y. Plotnikov

Reviewer 1:

1) I would give more prominence to the role of ketone bodies in the heart in section 2 as they may represent a benefit in some structural disease such as heart failure. In fact, ketone bodies represent a good alternative substrate, able to improve the cardiac metabolic efficiency. Some studies on humans and animal models have shown an improvement of the cardiac function and metabolism by beta-hydroxybutyrate (β-OHB), thus inducing reverse ventricular remodelling, with a consequent improvement in the cardiac output and diastolic function. (doi: 10.3390/ijms22115863).

Thank you for your appreciation of our manuscript.

We have added the paragraph about the role of ketone bodies in the heart failure with related references:

Nielsen, R.; Møller, N.; Gormsen, L.C.; Tolbod, L.P.; Hansson, N.H.; Sorensen, J.; Harms, H.J.; Frøkiær, J.; Eiskjaer, H.; Jespersen, N.R.; Mellemkjaer, S.; Lassen, T.R.; Pryds, K.; Bøtker, H.E.; Wiggers, H. Cardiovascular Effects of Treatment With the Ketone Body 3-Hydroxybutyrate in Chronic Heart Failure Patients. Circulation 2019, 139, 2129-2141.

Palmiero, G.; Cesaro, A.; Vetrano, E.; Pafundi, P.C.; Galiero, R.; Caturano, A.; Moscarella, E.; Gragnano, F.; Salvatore, T.; Rinaldi, L.; Calabrò, P.; Sasso, F.C. Impact of SGLT2 Inhibitors on Heart Failure: From Pathophysiology to Clinical Effects. Int. J. Mol. Sci. 2021, 22, 5863.

Rajeev, S.P.; Wilding, J.P. SGLT2 inhibition and ketoacidosis—should we be concerned? Br. J. Diabetes Vasc. Dis. 2015, 15, 155–158.

Santos-Gallego, C.G.; Requena-Ibanez, J.A.; Antonio, R.S.; Ishikawa, K.; Watanabe, S.; Picatoste, B.; Flores, E.; Garcia-Ropero, A.; Sanz, J.; Hajjar, R.J.; et al. Empagliflozin Ameliorates Adverse Left Ventricular Remodeling in Nondiabetic Heart Failure by Enhancing Myocardial Energetics. J. Am. Coll. Cardiol. 2019, 73, 1931–1944.

Reviewer 2 Report

The manuscript submitted to Nutrients by Yakupova et al., titled: "Effects of ketogenic diet on muscle metabolism in health and disease" is a review of the evidence investigating the relationship between ketogenic diet and muscle metabolism in various diseases as well as in health state.  The manuscript is well organized and structured, the methodology is appropriate, and it provides a set of valuable results.

Author Response

Dear Editors, 

Thank you for giving us a chance to revise our manuscript. We appreciate the work of the editor and reviewers and express great gratitude for the positive assessment of our article. We have modified the text to answer all recommendations expressed in the reviewer’s comments. We have made the appropriate changes in the text.

Thank you for your work.

Below, we present the reviewer’s specific comments (in black) with our replies (in blue).

Sincerely,

Elmira I. Yakupova,
Egor Y. Plotnikov,

Reviewer 2:

1) The manuscript submitted to Nutrients by Yakupova et al., titled: "Effects of ketogenic diet on muscle metabolism in health and disease" is a review of the evidence investigating the relationship between ketogenic diet and muscle metabolism in various diseases as well as in health state.  The manuscript is well organized and structured, the methodology is appropriate, and it provides a set of valuable results.

Thank you for your appreciation of our manuscript!